# Reproducibility study of "FairLISA: Fair User Modeling with Limited Sensitive Attributes Information"

## Abstract

This is a reproducibility study of the paper "FairLISA: Fair User Modeling with Limited Sensitive Attributes Information" by Zhang et al. (2023). It proposes a method of increasing fairness in user modeling tasks, by filtering out sensitive information from user embeddings. In contrast to other fairness aware methods, FairLISA is designed for filtering data with both known and unknown sensitive attributes. In this paper we explain the method from the paper, the claims about the effectiveness of the method, and our process of attempting to recreate said claims. We test the reproducibility of their original claims, test the generalisability of their method, and provide our implementation of the FairLISA method so further research can be done. We conclude that none of the claims of the original paper are fully reproducible in a reasonable amount of time. Some of the claims were able to be partially reproduced, and we detail those results.

## 1 Introduction

The growing influence of neural networks on our society calls for the development of techniques in which affected parties are protected from possible harm. To this end, it is vital to develop transparent (Arrieta et al., 2020) neural networks that process user data fairly and confidentially (Kearns & Roth, 2019) while keeping a clear indication of accountability (Kim & Doshi-Velez, 2021) should problems arise. A widely-used application of neural networks where fairness plays an important role is user modeling. User modeling is the act of profiling latent characteristics of users from known information or behaviour. Whereas these latent characteristics are crucial to making accurate predictions related to the user, they can capture unfair and harmful biases related to sensitive attributes of the user (Pessach & Shmueli, 2022).

To prevent harmful biases from affecting decision-making, a common approach is to hide sensitive attributes from the network so that it does not train on them (Wu et al., 2021). This approach, however, assumes that the labels for the sensitive attributes are available to the developer, which is not always the case in practical applications. To ensure fair treatment of user data in a practical setting, FairLISA is proposed. FairLISA is a new user modeling approach in which an adversarial neural network is trained on both users whose sensitive information is available and unavailable (Zhang et al., 2023). In the context of cognitive diagnosis and recommender systems, FairLISA appears to outperform existing methods such as FairGO (Wu et al., 2021) and FairGNN (Dai & Wang, 2021).

In this paper, we will reproduce the results presented in the FairLISA paper and validate the central claims made by the authors. Additionally, we describe our process in obtaining said results.

## 2 Scope of reproducibility

In their paper Zhang et al. (2023) propose the new method FairLISA, a general framework for _Fair user modeling with LImited Sensitive Attributes_. This method should ensure further fairness in user modeling tasks, by filtering out sensitive information from user embeddings. How it differs from other fairness-aware methods, is in taking a lack of specified or known sensitive attributes in the data into account and still being able to improve fairness.

The authors make the following claims in regards to their new approach:

1. **Outperforms other fairness-aware baselines in limited sensitive information situations.** When comparing FairLISA to other methods (ComFair (Bose & Hamilton, 2019), FairGo (Wu et al., 2021), FairGNN (Dai & Wang, 2021)) the authors claim their method achieves better scores in terms of fairness, with less of a trade-off in terms of accuracy. This is desirable since an increase in fairness usually results in a loss of accuracy (Menon & Williamson, 2018). FairLISA should also perform significantly better than models that are not designed to work in limited sensitive information situations, where not all sensitive features in the data are known, such as ComFair and FairGo.

2. **Achieves more robust results on different missing ratios.** Results in the paper show that the more sensitive attributes are missing from the data, the worse fairness-aware methods perform. Since FairLISA is developed to deal with missing sensitive features, however, it should perform better than other fairness-aware methods, even when compared to other methods that also take missing features into account, such as FairGNN.

3. **Including data with known and unknown sensitive attributes contributes to the fairness results.** A novel element of the FairLISA method using both data with and without sensitive features to improve their filter. The authors investigate the impact of data with and without sensitive attributes by adjusting corresponding hyperparameters. They find that while data with sensitive features has a greater impact, both types of data improve fairness.

4. **Outperforms other fairness-aware baselines on several classical group fairness metrics.** The authors claim that FairLISA also performs well on classical group fairness metrics Demographic Parity (Dwork et al., 2012) and Equal Opportunity (Hardt et al., 2016), and performs better compared to other methods.

Of these claims we were able reproduce parts of the first three claims to varying degrees. The fourth claim we were not able to reproduce due to time constraints. Other than reproducing these claims, we also tried to show further generalisability of this method by using the method on a dataset other than the ones mentioned in the paper. Lastly, we reorganised the code to improve readability and ease of running experiments.

## 3 Methodology

The authors provided a codebase which was, however, non-functional and incomplete, with parts of the implementations contradicting the paper. With the provided code as a baseline, we have revised and expanded the code, including support for the pre-processed Movielens-1M and COMPAS datasets as well as the missing PMF and NCF algorithms. We have refactored the code and provided comments for increased clarity and efficiency. In addition to refactoring the provided FairLISA framework, we have written scripts to pre-process the datasets.

### 3.1 Description of the FairLISA method

The FairLISA method consists of training a Filter, $\mathcal{F}$, to filter the given user embeddings, $\theta$, and remove sensitive attributes before passing the filtered embeddings, $\hat{\theta}$, through a model. The performance of this Filter $\mathcal{F}$ is tested and improved by the presence of a Discriminator, $\mathcal{D}$, for each of the $n$ sensitive features present in the data. This Discriminator $\mathcal{D}$ tries to guess the value of a given sensitive attribute based on the filtered user embeddings $\hat{\theta}$. If the Discriminator is not able to guess the value of a given sensitive attribute, this means that the Filter has successfully filtered sensitive information from the user embeddings. The FairLISA method specifically utilises user embeddings that are derived from data with both known and and unknown sensitive attributes.

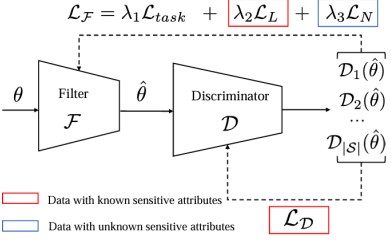

Figure 1: FairLISA architecture

In detail the method is as follows: the Filter $\mathcal{F}$ is trained by fixing the Discriminators $\mathcal{D}_n$, the discriminators for each sensitive attribute $n$, and minimising the filter loss, $\mathcal{L}_\mathcal{F}$, which is shown in equation 1. Afterwards, the Filter $\mathcal{F}$ is fixed and the Discriminators $\mathcal{D}_n$ are trained by minimising the discriminator loss, $\mathcal{L}_\mathcal{D}$, as given in equation 4. This adversarial training process is illustrated in Figure 1.

$$\mathcal{L}_\mathcal{F} = \lambda_1 \mathcal{L}_{task} + \lambda_2 \mathcal{L}_L + \lambda_3 \mathcal{L}_N \tag{1}$$

As shown in equation 1, the filter loss is made up of three different losses. First is the task loss, $\mathcal{L}_{task}$. The formula for calculating this loss is dependent on the model that is being trained. For more details on the models the filter is applied to, see section 3.1.1. The second loss, $\mathcal{L}_L$, is the loss of the dataset with known sensitive attributes, $U_L$, the formula for this loss is equation 2. The third loss is $\mathcal{L}_N$, the loss of the dataset with unknown sensitive attributes, $U_N$. The formula for this loss is equation 3. $\lambda_1$, $\lambda_2$, and $\lambda_3$ are hyperparameters that modify the weight of the three losses in the final filter loss formula.

In the following equations, these elements refer to the following: $\hat{\theta}_i$ is the filtered representation of user embedding $u_i$. $l$ is the number of users in the training data that have known sensitive attributes. $n$ is the number of users that have their sensitive attributes removed from the training data. $s$ is one of the sensitive features the method tries to filter out.

$$\mathcal{L}_L = \frac{1}{l} \sum_s \sum_{1 \leq i \leq l : s_i = s} \log \mathcal{D}_s(\hat{\theta}_i) \tag{2} \qquad \mathcal{L}_N = \frac{1}{n} \sum_{l+1 \leq i \leq l+n} \sum_s \mathcal{D}_s(\hat{\theta}_i) \log \mathcal{D}_s(\hat{\theta}_i) \tag{3}$$

$$\mathcal{L}_\mathcal{D} = -\frac{1}{l} \sum_s \sum_{1 \leq i \leq l : s_i = s} \log \mathcal{D}_s(\hat{\theta}_i) \tag{4}$$

### 3.1.1 Applied models

Since the FairLISA method is a filter that needs to be applied to models with user modeling tasks, the paper tests their filter for six models. Three Cognitive Diagnosis models use the Pisa2015 dataset and three recommender systems which use the MovieLens-1M dataset. More details on these datasets can be found in section 3.2.

**Cognitive diagnosis models**

These models aim to assess whether an individual has mastered a certain set of skills or abilities (Templin & Henson, 2006). The IRT, MIRT and NCD models are cognitive diagnosis models that are specifically applicable to the field of education.

- **IRT**, Item Response Theory, is a unidimensional method that uses a logistic-like function to model students' latent traits based on test scores (Lord, 1952).

- **MIRT**, Multidimensional Item Response Theory, is generally an extension of the basic IRT method but uses multidimensional vectors to characterise students (Reckase, 2009), (Chalmers, 2012).

- **NCD**, Neural Cognitive Diagnosis, uses a deep neural network to model students (Wang et al., 2020).

Implementations of these models were present in the code provided by the authors, so we could expand those implementations to work in a more general manner.

**Recommender systems**

These models give recommendations or predict preferences based on modeled user preferences. These specific instances of recommender systems are also called collaborative filtering (Su & Khoshgoftaar, 2009).

- **PMF**, Probabilistic Matrix Factorization, is a collaborative filtering approach that uses latent user and item matrices over a Gaussian prior distribution to make predictions (Mnih & Salakhutdinov, 2007).

- **NCF**, Neural network Collaborative Filtering, uses a deep neural network to make recommendations (He et al., 2017).

- **LightGCN**, is a model based on a simplified Graph Convolution Network that is optimised for collaborative filtering (He et al., 2020).

The implementations of these models were not present in the code provided by the authors and no further instruction was given by the paper. We attempted to recreate the models using online repositories of PMF[1] and NCF[2].

## 3.2 Datasets

The original paper makes use of two datasets: Pisa2015 and MovieLens-1M. We also conducted further experiments with COMPAS. General information about these datasets can be found in Table 1.

Table 1: General information for Pisa2015, Movielens-1M and COMPAS

| Task | Dataset | Train/Test/Validation | Sensitive features | Link |
|------|---------|----------------------|--------------------|------|
| Cognitive Diagnosis | Pisa2015 | 1999k/571k/286k | Gender, Region, Family Education, Family Economic | Google Drive |
| Movie Recommendation | Movielens-1M | 704,6k/198,6k/97,0k | Gender, Age, Occupation | Grouplens |
| Recidivism risk analysis | COMPAS | 42,5k/12,2k/6,1k | Gender, Ethnicity, Marital status | GitHub |

### 3.2.1 PISA2015

The Programme for International Student Assessment (PISA) is a questionnaire that aims to quantify students' reading, mathematics, and scientific knowledge. In the 2015 version of this dataset, there are 471919 students for which we have data for the sensitive attributes as well as answers to at least some of the 183 questions. This dataset was obtained by combining the financial literacy data from OECD [3] with a pre-processed version of the PISA2015 student questionnaire as obtained from (bigdata ustc, 2021).

The result is a dataset where the entries are presented as in Table 2. The sensitive attributes used from this dataset are gender value of home possessions, region, and the highest. To make sure there is enough data, only students who answered at least 20 questions were kept. Due to computational constraints, from this dataset 100,000 students were randomly selected and split into a train, test and validation split with ratio 7:2:1 as seen in Table 1.

Table 2: Example entries for the pre-processed Pisa2015 dataset

| user_id | item_id | score | gender_split | region_split | edu_split | home_split |
|---------|---------|-------|--------------|--------------|-----------|------------|
| 39812 | 148 | 0 | 0 | 1 | 1 | 0 |
| 45586 | 46 | 1 | 0 | 1 | 1 | 1 |
| 14264 | 152 | 0 | 0 | 1 | 1 | 0 |
| 41896 | 112 | 1 | 1 | 1 | 1 | 0 |
| 11942 | 134 | 0 | 1 | 1 | 0 | 1 |

### 3.2.2 MovieLens-1M

MovieLens-1M is a dataset comprised of 1,000,209 anonymous ratings of approximately 3,900 movies made by 6,040 MovieLens users who joined MovieLens in 2000. For the Movielens-1M dataset, the raw .dat files have been taken from the official source. Using a custom script, the three data files from the dataset are

---

[1] https://github.com/xuChenSJTU/PMF-Pytorch/tree/master
[2] https://github.com/guoyang9/NCF
[3] https://www.oecd.org/pisa/data/2015database/

merged into a single dataset containing all rating entries for all users. The full dataset is split into the train, test, and validation sets which are structured in the format as seen in Table 3.

Table 3: Example entries for the pre-processed MovieLens-1M dataset

| user_id | gender | age | occupation | zip-code | item_id | title | genres | score | timestamp |
|---|---|---|---|---|---|---|---|---|---|
| 0 | 0 | 0 | 1 | 48067 | 1192 | One Flew Over the Cuckoo's Nest (1975) | Drama | 5 | 978300760 |
| 32 | 1 | 0 | 0 | 55421 | 1096 | E.T. the Extra-Terrestrial (1982) | Children's|Drama|Fantasy|Sci-Fi | 4 | 978982390 |
| 2454 | 1 | 1 | 1 | 48197 | 1269 | Back to the Future (1985) | Comedy|Sci-Fi | 1 | 974180325 |

### 3.2.3 COMPAS

COMPAS is a recidivism risk assessment dataset which contains personal information and risk scores for 18,610 individuals from the United States. The data from the COMPAS dataset contains personal information that can be used as sensitive attributes. We have chosen to include gender, ethnicity, and marital status as the sensitive attributes for this dataset. The three chosen attributes are converted to binary values to match the format of the pre-processed Pisa2015 dataset. For ethnicity, a binary distinction is made between those whose ethnicity is registered as Caucasian and non-Caucasian. For marital status, a distinction is made between individuals who are married or have a significant other and those who do not. An example of the pre-processed data can be seen in Table 4.

Table 4: Example entries for the pre-processed COMPAS dataset

| gender | ethnic | marital | user_id | item_id | score |
|---|---|---|---|---|---|
| 1 | 1 | 1 | 13004 | 1 | 0 |
| 0 | 0 | 1 | 2208 | 2 | 1 |
| 1 | 0 | 0 | 15529 | 0 | 1 |

### 3.3 Hyperparameters

There are three hyperparameters described in the paper. $\lambda_1$, $\lambda_2$, and $\lambda_3$ are used in the loss function used to train the sensitive filter, see 1. They adjust the task loss of the used model, the loss of the known feature data ($U_L$), and the loss of the unknown data ($U_N$), respectively. The paper sets the values of $\lambda_1$, $\lambda_2$, and $\lambda_3$ to 1, 2, and 1 respectively for cognitive diagnosis and 1, 20, and 10 for recommender systems. In claim 3 different values for these hyperparameters are tested.

In the code base provided by the authors, the values of $\lambda_2$ and $\lambda_3$ seems to be 1 and 0.5, possibly due to the results of claim 3. The learning rate is set to 0.001, except for the learning rate of the discriminator, which is set to 0.01. There is a hyperparameter that sets the missing ratio of sensitive attributes from the data. This is 0.2 or 20%. Different values for missing ratios are also tested in claim 2. Unless stated otherwise, these are the hyperparameter values we ran our experiments with.

### 3.4 Experimental setup and code

An anonymous github repository with our refactored FairLISA code, added code and command lines to run our experiments is available at https://anonymous.4open.science/r/fairlisa-reproduction/ and has been added as a supplementary resource.

Firstly, we use the following metrics for our results. The effectiveness of the sensitive filter is tested by training an attacker with the same architecture as the discriminator. This attacker tries to guess the value of a given sensitive attribute based on the filtered embeddings. The performance of this attacker is evaluated by an auc score. This metric signifies the area under the curve and the lower it is, the fairer the results. The other testing metric is for testing the accuracy of the given models. RMSE, or regularised mean squared error, the lower, the more accurate the model is.

For claim 1 we ran the code for the Pisa2015 dataset with the refactored implementations of the IRT, MIRT and NCD models. We trained the filter for 100 epochs, the discriminator for 10 epochs and the attacker

for 10 epochs, with the standard hyperparameters provided in the code. The MovieLens-1M dataset on our implementations of PMF and NCF was trained for 10 epochs on the filter, 10 on the discriminator and 10 on the attacker, also with the standard hyperparameters. The COMPAS dataset on PMF and NCF was trained for 100 epochs on the filter, 10 on the discriminator and 10 on the attacker, with the standard hyperparameters.

For claim 2 we trained the filter for 50 epochs, the discriminator for 10 and the attacker for 10 epochs. We did this for both the NCD and NCF models, for the missing ratios of 0.2, 0.4, 0.4, 0.8 and 0.95.

For claim 3 we used the same amount of epochs as for claim 2 and varied the hyperparameters args.FAIRNESS_RATIO for $\lambda_2$ and args.FAIRNESS_RATIO_NOFEATURE for $\lambda_3$, with the values 0, 0.5, 1 and 1.5.

In the paper Demographic Parity and Equal Opportunity are also used to validate claim 4, since this claim has not been reproduced, these metrics have not been included here.

### 3.5 Computational requirements

For the recommender system tests on the Movielens-1M dataset for PMF and NCF, a single NVIDIA GeForce RTX 2070 SUPER with 8GB VRAM was used. A total of 13.9 GPU hours were needed to run the experiments on the Movielens-1M dataset. For the COMPAS dataset, the same hardware setup was used. For the cognitive diagnosis models NCD and MIRT an A100 PCIe 40/80Gb was used and roughly a total of 70 GPU hours were needed to run the experiments on the Pisa2015 dataset. The IRT experiments were ran on a RTX 3050 for 20 hours.

## 4 Results

**Claim 1: Outperforms other fairness-aware baselines in limited sensitive information situations**

*Partially Reproduced*

We were able to partially reproduce the values provided in the FairLISA paper with our implementation of the code, which can be seen in Table 6 and Table 5. Due to reasons further discussed in section 5, we did not succeed in a full reproduction of the results.

Table 5: AUC scores for the sensitive features and RMSE for the Movielens-1M dataset using PMF and NCF. G, A, and O stand for gender, age, and occupation respectively. Green is used to indicate a higher performance on the reproduction, while red is used to denote a lower performance. Orange results indicate inconclusive results. Results that outperform the origin are written in bold.

|  | PMF | | | | NCF | | | |
|---|---|---|---|---|---|---|---|---|
|  | AUC-G | AUC-A | AUC-O | RMSE | AUC-G | AUC-A | AUC-O | RMSE |
| Origin | 0.6862 | 0.7235 | 0.6656 | 0.8670 | 0.6915 | 0.7153 | 0.6625 | 0.8635 |
| FairLISA paper | 0.5193 | 0.5240 | 0.5210 | 0.8935 | 0.5281 | 0.5220 | 0.5222 | 0.8853 |
| FairLISA reproduction | **0.5136** | 0.5 | 0.5 | 0.8960 | **0.5580** | 0.5 | 0.5 | 0.8924 |

Table 6: AUC scores for the sensitive features and RMSE for the Pisa2015 dataset using IRT, MIRT and NCD. R, A, E, and C stand for region, age, parental education level, and family economic status respectively. Green is used to indicate a higher performance on the reproduction, while red is used to denote a lower performance. Results that outperform the origin are written in bold.

|  | IRT | | | | | MIRT | | | | | NCD | | | | |
|---|---|---|---|---|---|---|---|---|---|---|---|---|---|---|---|
|  | AUC-R | AUC-A | AUC-E | AUC-C | RMSE | AUC-R | AUC-A | AUC-E | AUC-C | RMSE | AUC-R | AUC-A | AUC-E | AUC-C | RMSE |
| Origin paper | 0.5864 | 0.5191 | 0.5930 | 0.5565 | 0.2042 | 0.6271 | 0.5351 | 0.6329 | 0.5831 | 0.1903 | 0.6307 | 0.5344 | 0.6327 | 0.5808 | 0.1704 |
| FairLISA paper | 0.5412 | 0.5163 | 0.5490 | 0.5279 | 0.2044 | 0.5540 | 0.5181 | 0.5952 | 0.5429 | 0.1896 | 0.5540 | 0.5154 | 0.5727 | 0.5429 | 0.1796 |
| FairLISA reproduction | 0.5921 | 0.5201 | 0.5979 | 0.6191 | 0.1992 | **0.5793** | 0.5434 | **0.5751** | **0.5834** | 0.2477 | **0.6167** | 0.5474 | **0.6032** | 0.6315 | 0.2271 |

**Claim 2: Achieves more robust results on different missing ratios**

*Partially Reproduced*

In Figure 2 the results for the missing ratios for the NCD and NCF on the sensitive attributes Region, Gender, Age and Gender are shown. Our results partially align with those described in the FairLISA paper.

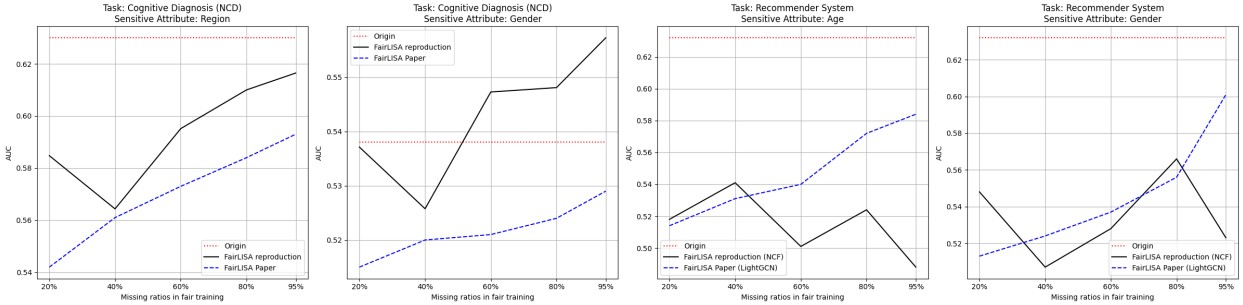

Figure 2: Fairness performance on different missing sensitive attributes ratios (the lower, the fairer)

**Claim 3: Including data with known and unknown sensitive attributes contributes to the fairness results**

*Partially Reproduced*

As seen in Figure 3 we were able to partially reproduce the trends shown in the original FairLISA paper and only for the NCD model.

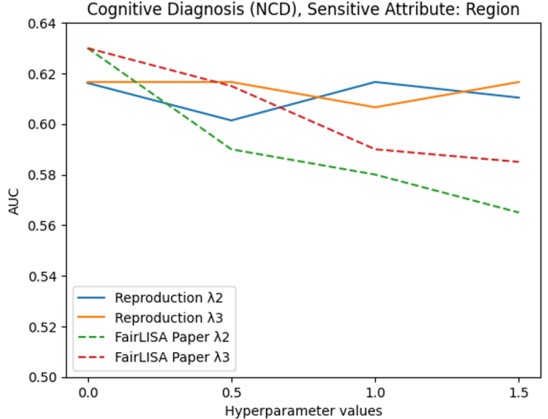 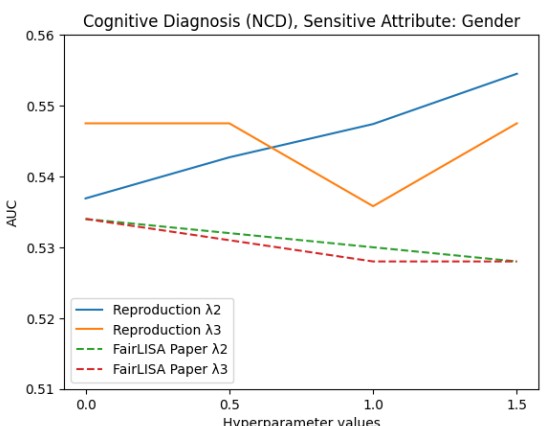

(a) Paper and reproduction AUC scores for the region feature as the values lambda-2 and lambda-3 are individually increased.

(b) Paper and reproduction AUC scores for the gender feature as the values lambda-2 and lambda-3 are individually increased.

Figure 3: Fairness performance for different hyperparameter values

**Claim 4: Outperforms other fairness-aware baselines on several classical group fairness metrics**

*Not Reproduced*

The paper includes a description of the implementation of the Demographic Parity and Equal Opportunity metrics in the appendix, but this was not implemented in the provided codebase.

**Extension on the original paper**

We also tested the performance of the COMPAS dataset on the FairLISA framework. In Table 7, the results for the PMF and NCF models are shown

Table 7: AUC scores for the sensitive features for the COMPAS dataset using PMF and NCF. G, E and M stand for gender, ethnicity and marital status respectively.

| | PMF | | | | NCF | | | |
|---|---|---|---|---|---|---|---|---|
| | AUC-G | AUC-E | AUC-M | RMSE | AUC-G | AUC-E | AUC-M | RMSE |
| Origin | 0.4922 | 0.5019 | 0.5021 | 0.0754 | 0.4951 | 0.5083 | 0.5114 | 0.0391 |
| FairLISA | 0.5377 | 0.5863 | 0.6594 | 0.7717 | 0.5382 | 0.5859 | 0.6548 | 0.7766 |

# 5 Discussion

**Claim 1**

For the Pisa2015 dataset, the reproduced results for running MIRT and NCD follow a similar trend compared to the results in the paper. The results, however, are noticeably lower compared to the paper results. the AUC-A score in particular, performs worse compared to the baseline from the paper.

For PMF and NCF, FairLISA appears to produce results that are comparable to the results shown in the paper. In particular, AUC-R and RMSE for PMF and NCF are close to the original values from the paper. For AUC-A and AUC-O however, our results produce an unusual AUC score of exactly 0.5, indicating that the attacker is not able to perform beyond random guessing. These AUC values for age and occupation are likely caused by our choice of using a binary threshold to calculate the AUC scores. The paper does not elaborate on how the roc_auc_score has been calculated for features that are not naturally binary. We have attempted both binarising the results before training and binarising the predicted feature values. For binarising the results ahead of time, however, the AUC for the age and occupation feature is less than 0.5. Thus, we opted to binarise the predicted features and compare them to the true binarised features. We have opted to not implement LightGCN as its unique structure would have taken a significant amount of time to integrate into the FairLISA framework.

**Claim 2**

The authors state that FairLISA and FairLISA+FairGO produce more robust results in situations where a high percentage of sensitive features are missing. As seen in the tables from our partial reproduction of claim 2, the "Age" sensitive attribute of the Movielens-1M dataset using NCF does not show a similar trend to the results presented in the paper. The "Gender" sensitive attribute partially shows a similar trend to the FairLISA results. One possible explanation might be that our data was trained for 10 filter epochs on the Movielens-1M dataset due to time constraints, which may have caused more noise to be present. For the Pisa2015 dataset using the NCD algortihm, reproduction results follow a similar upward trend to the results from the paper. The AUC scores however, are consistently higher compared to the paper results, indicating that the fairness of the reproduced results are is less compared to the paper results. For the "Gender" feature, an increase in missing ratios result in the reproduced AUC scores to exceed those of the origin presented in the FairLISA paper. These results can possibly be attributed to noise in the reproduced results or to differences in the pre-processing of the data.

**Claim 3**

For FairLISA, both data with known and unknown sensitive attributes should contribute to the fairness results. In the graph of the results of claim 3, a partial reproduction of this claim can be observed. The increase of lambda 2 and lambda 3 determines the weight of the respective loss terms for known and unknown data. An increase in the values for lambda 2 and lambda 3 should result in a lower AUC score, thus an increased fairness. In our reproduction efforts, however, an increase in the value for lambda 2 does not seem to reduce AUC scores. Lambda 3, appears to decrease slightly as its value increases. The decrease is noticeably less pronounced compared to the results shown in the paper. Lambda 2 dictates the weight of the loss of the users whose sensitive features are known. Thus, a more pronounced decrease in the AUC score is expected for lambda 2 compared to lambda 3. However, we have not found a similar behavior in

our reproduction of this experiment. Similarly, for the hyperparameter graph of the NCD "Gender" feature, we have found no decrease in AUC scores as the lambda values increase.

**Claim 4**

In the FairLISA paper, the authors state that FairLISA and FairLISA+FairGO generally outperform FairGO, ComFAIR, and the FairGNN based on Demographic Parity and Equal Opportunity. As the authors did not provide an implementation of these tests, we were unable to validate this claim within the allotted time.

**Extension**

For the COMPAS dataset, FairLISA produces results that follow similar trends to the results of the other datasets. The reason these results are accurate to the results presented in the paper is that we used a binary data format similar to Pisa2015 to train the FairLISA architecture. For the COMPAS dataset result, the origin baseline has been computed by using the original user embeddings from PMF and NCF without applying the sensitive filter. Our results, however, do not follow the same trend as the original values for the Pisa2015 and Movielens-1M datasets. One possible explanation for these results might be that the provided FairLISA framework has not been properly integrated with PMF and NCF.

## 5.1 What was easy

The paper is structured and the writing style does not use many acronyms or much field-specific language, making the core ideas presented in the paper relatively easy to comprehend. Additionally, the central research questions in the paper are stated clearly and structurally answered. In the code, implementations of IRT and MIRT are provided by the authors in the BaseCD.py file. The implementation made it possible to analyse the workings of these methods in the limited provided time.

## 5.2 What was difficult

**Datasets**

For the Pisa2015 and Movielens-1M datasets, the authors have pre-processed the data to match the FairLISA framework. No information on the pre-processing steps was provided. Especially Pisa2015 has been heavily pre-processed, which required us to apply a trial-and-error approach to find the correct data structure. In addition to the pre-processed data, the NCD model makes use of an item2knowledge.pkl file, the contents of this file were unknown until a clarification was provided by the authors. Lastly, separate attacker train and test sets are used alongside the discriminator test, train, and validation data. The contents of these files remain unclear to us.

**Code**

The code provided by the authors was hard-coded to run on the Pisa2015 dataset using the IRT, MIRT, and NCD models. The implementation for NCF, PMF, and LightGCN had not been provided, requiring us to integrate these algorithms into the FairLISA framework ourselves. Another challenging aspect was the quality of the provided code. The code lacks comments and contains unusual structures such as if-statements that are never true and the validation set that is not used. These structures have made the code difficult for us to interpret. Furthermore, the code contains implementations for two filter modes, which are not elaborated on in the code.

**Paper**

The paper and code have shown discrepancies in their contents. The provided implementations of the Discriminator and Filter, as well as the usage of hyperparameters, do not match the paper. In the paper, the baseline methods of ComFair, FairGNN, and FairGO have been used. No specifications are given on how the authors obtained the results for these methods using the cognitive diagnosis and recommender system models. Lastly, no details are given on how FairGO and FairLISA are combined to obtain the results for FairGO+FairLISA.

### 5.3 Environmental impact

Training a neural network usually is an energy-intensive process. The experiments on IRT for the Pisa2015 dataset have been run on a private infrastructure using a GTX 3050 graphics card for 20 hours. For CO2 emission calculations, we use the mlco2 github page with a carbon efficiency of 0.350 mlco2. Using the GTX 3080 as a similar graphics card to the Nvidia GTX 3050 to compute the expected pollution for training the model, we obtain a value of roughly 2.2 kg CO2 emitted to run these experiments. For the MIRT and NCD models, we have used the Snellius computer cluster from SURF. We have run the models for MIRT and NCD using an Nvidia A100 graphics card for a total of 70 hours. An estimated 6.1 kg CO2 is emitted through the training of these models. Lastly, to run the experiments we conducted on the recommender system models, a total of 16.9 GPU hours are needed using an Nvidia RTX 2070 Super. Using an Nvidia RTX 2080 as an equivalent graphics card for pollution calculations, we estimate an average of 1.3 kg CO2 has been emitted through training the recommender system models. For all experiments combined, the pollution is equivalent to emitting 9.6 kg CO2 into the atmosphere.

### 5.4 Communication with original authors

Communication with the authors of the FairLISA paper has been limited to an e-mail in which a clarification on the Pisa2015 format and item2knowledge.pkl file has been requested. The authors have provided an example of a single user in their pre-processed Pisa2015 dataset and its corresponding item2knowledge file. A follow-up e-mail has been sent to the authors to request further examples of their implementation of the PMF, NCF, and LightGCN algorithms as these algorithms are not available in the GitHub repository associated with the paper. The authors have not replied to the second e-mail.

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
