# OpenReview forum: "Reproducibility study of "FairLISA: Fair User Modeling with Limited Sensitive Attributes Information""
_TMLR — Rejected by TMLR_

### Review · Reviewer_FzpV · 2024-03-22

**Summary Of Contributions:**

The paper is concerned with reproducing the experimental result of Zhang et al. Four claims are listed in the paper and they are verified by reproducing the experimental results. The paper gives a detailed description of the datasets and the experimental procedure that was followed highlighting seem reproducibility difficulties from Zhang et al.

**Audience:**

No

**Broader Impact Concerns:**

I don't think that there are broader impact concerns.

**Claims And Evidence:**

Yes

**Requested Changes:**

-It would be better to include some facts  about the conclusions of the paper in the abstract.

-I think it’s better to replace “limited sensitive situations" → “limited sensitive information situations"

-Typo on page 4: ”no further instruction was given paper" → “no further instruction was given by the paper”

**Strengths And Weaknesses:**

Strengths:
-It is certainly important for the community to give reproducible empirical results and the paper contributes in this direction.


Weaknesses:
W1-Unfortunately, I don't find the paper interesting. First, it would be more interesting to reproduce a seminal paper that has had more impact. I don't work on this problem specifically, but why Zhang et al, in particular.  Second, although there are some differences in the final experimental results from the original paper, they do not appear to be very significant.

W2-For claim 4, did this paper produce the code based on the description given in the appendix?

W3-The code is very likely to have some randomness/random seed. How did this paper handle this?

---

> ### Author Response · Authors · 2024-04-16
> **Implemented requested changes**
>
> I have made a revision according to your requested changes, would this be about what you had in mind?
>
> As for the questions stated under strengths and weaknesses:
> No, we did not produce the code for claim 4 based on the description given in the appendix of the paper.
> In regards to randomness, the original codebase provided by the authors did include a -SEED argument with a default value, so in our code and for our experiments we use the same argument with the given default value.

---

> > ### Comment · Reviewer_43XR · 2024-04-19
> > **Other reviews?**
> >
> > Dear authors, did you look at the other reviews and do you have any questions or comments about the requested changes?

---

> > > ### Author Response · Authors · 2024-04-23
> > >
> > > Our apologies; not yet no. We will look at the other reviews and see if there are any questions or comments. At a cursory glance I have some questions regarding the specifics of the requested changes, but I will leave those on the specific review in question. Our apologies once again for the slow response.

---

> > > > ### Comment · Action_Editor_c761 · 2024-04-25
> > > >
> > > > Dear authors,
> > > >
> > > > We have already well passed the standard duration of the discussion phase (2 weeks after receiving all reviews). Please do hurry up if you would like to follow up with the other reviews, as reviewers are already entering their decision recommendations.
> > > >
> > > > Thank you

---

### Review · Reviewer_Ymv8 · 2024-03-25

**Summary Of Contributions:**

The paper reproduces the results from a recently accepted NeurIPS paper and finds that some of the reported results cannot be replicated.

**Audience:**

Yes

**Claims And Evidence:**

No

**Requested Changes:**

I would like to see an exploration of different choices for parameters and how they affect the results.

**Strengths And Weaknesses:**

Strength:

-- The paper addresses an important problem. Replicability is a big issue in scientific research and highlighting applicability issues is somewhat under-appreciated in our community.

Weaknesses:

-- It is hard to assess the claim of the papers. It seems like the original paper has some parameters and the choice of those parameters is not well described in the original paper/code. The paper does not experiment with different choices of these parameters and their effect on the results.

-- Without being familiar with the code and the ability to rerun the experiments, it is virtually impossible to verify the claim of the papers.

---

> ### Author Response · Authors · 2024-04-26
>
> Dear reviewer, thank you for your review. Our apologies for the late response.
>
> What parameters are you referring to specifically? We do experiment with different values for different parameters, such as the ratio of how much of the data has known or unknown sensitive attributes, or how heavily the known or unknown data is weighted in the loss function used for training the filter. It's unfortunately not clear to me what you are specifically requesting.

---

> > ### Comment · Reviewer_Ymv8 · 2024-04-26
> > **Response**
> >
> > $\lambda_1$, $\lambda_2$ and $\lambda_3$. You mention "In the code base provided by the authors, the values of $\lambda_2$ and $\lambda_2$ seems to be 1 and 0.5, possibly due to the results of claim 3." Without knowing exactly what parameters are used by other paper, it would be hard to replicate their result.

---

> > > ### Author Response · Authors · 2024-04-26
> > >
> > > Ah I see, it is true that it is unclear from the paper which values for $\lambda_1$, $\lambda_2$, and $\lambda_3$ they ultimately decide is best to use. But in terms of reproducibility, they do vary the values for $\lambda_2$, and $\lambda_3$ in claim 3 and give the different parameter values they vary over. So for claim 3 at least we did try different values of $\lambda_2$, and $\lambda_3$ for the NCD model, based on the values specified in the paper:
> > > 'Specifically, we vary the values of $\lambda_2$, $\lambda_3$ on NCD as {0,0.5,1,1.5}'
> > >
> > > For the rest of the experiments we keep the parameter values as provided in the codebase attached to the original paper, which had the default values of 1 and 0.5, but instead of $\lambda_2$ and $\lambda_3$, they're called args.FAIRNESS_RATIO and args.FAIRNESS_RATIO_NOFEATURE.

---

### Review · Reviewer_43XR · 2024-04-03

**Summary Of Contributions:**

This paper proposes a reproduction of the experimental part of a NeurIPS paper 2023 where the authors introduce an algorithm called "FairLISA" that proposes a method to increase fairess in classification algorithms. The paper proposes an open-source implementation of the algorithm and replicates the numerical results of the original paper. Part of the replication is successful (they obtain the same results in as the original paper) but not all.

**Audience:**

Yes

**Broader Impact Concerns:**

The paper replicates existing experiments. I do not think that a broader impact statement is needed.

**Claims And Evidence:**

Yes

**Requested Changes:**

- Clarify the contributions by analyzing the reproduction results.
- Clarify section 3.1.

**Strengths And Weaknesses:**

Strengths:
- Reproducing published paper is an interesting process that is not done enough.
- The paper point out some ambiguities in the original description of the algorithm and proposes solution to clarify it.
- The replication indicate that not all results were successfully reproduced.

Weaknesses:
- The results should be better analyzed and commented. In particular, the experimental part indicate that some results were successfully replicated but not all. These disparities should be clearly stated in the introduction: the introduction should state the contributions clearly and explain which results are not the same.
- Section 3.1 about model description is quite obscure for someone who did not already read the FairLISA paper. This section should I think be rewritten. Also, notations could be explained more clearly. For instance:
  - in (2): is it theta or \hat{\theta}
  - what are the indices "l" in equation (2) and (3)?
  - could the loss function be written in full form to be more precise?

About claims and evidence: I was not able to download the code on the anonymized repository. I do believe that the code is probably easy to run but I was not able to reproduce the experiments myself because of that. Hence, it is hard for me to judge if the "claims made in the submission supported by accurate, convincing and clear evidence" as requested by TMLR. I give the benefit of doubt to the author as the code seems to be there.

---

> ### Author Response · Authors · 2024-04-26
> **Clarifications requested changes**
>
> Dear reviewer, thank you so much for your review. Our apologies again for the late response.
>
> I had a couple of questions regarding the requested changes.
>
> Regarding the rewriting of the introduction to state our contributions and reproduction results. Do you mean that before the Scope of Reproducibility section or after the description of the claims in that section? I would assume after since otherwise the claims are not clearly defined, but I wanted to check whether this is what you intended.
>
> Regarding section 3.1:
> I can certainly rewrite it, but regarding the notations you mentioned in the strengths and weaknesses section, those are already explained in the paragraph below equations (2) and (3).
> - In (2) it is supposed to be \hat{\theta} yes, all the loss functions work with the filtered user embeddings.
> - The indices 'l' in (2) and (3) are 'the number of users in the training data that have known sensitive attributes'
> - Since the full loss function for the Filter includes the task specific loss, I would not be able to write that one in full form, since it depends on what model is being used. I could include the Filter loss function with  \mathcal{L}_L and \mathcal{L}_N written out, if you think that would improve readability?
> I can see if I can generally reorganise the 3.1 section to make it clearer where the notation on the different functions is explained.
>
> I hope you can answer these questions, if it is not too much trouble.

---

> > ### Comment · Reviewer_43XR · 2024-04-29
> > **Clarification of the writing**
> >
> > Dear authors,
> >
> > There are two independent comments:
> > 1. about the introduction: what I meant is that I think that the introduction should make clear what are the results of the paper. It is fine if you write a "scope of reproducibility" but I think that the introduction should mention what is successfully reproduced and what is not (after the scope)
> > 2. about section 3.1: if you can just clarify what you wrote in the paper, that should be OK.

---

> > > ### Author Response · Authors · 2024-04-30
> > >
> > > Dear reviewer, thank you for your response.
> > >
> > > I think I understand what you mean now, I have made a revision with the changes as requested. Is this what you intended?

---

### Decision · Action_Editor_c761 · 2024-05-28

**Recommendation:** Reject

**Comment:**

The authors may consider submitting a major revision at a later time, if they address the above limitations.

**Audience:**

The method studied in this paper is very recent, and it is not clear if it has attracted some attention. More importantly, the limitations of the study discussed above significantly reduce the interest of this paper for the TMLR audience.

**Claims And Evidence:**

This paper is an attempt at reproducing the results of a recently proposed approach for improving fairness in user modeling tasks. While reproducibility studies are in the scope of TMLR, this paper falls short of properly studying the robustness and generalizability of the method (as required by the [acceptance criteria of TMLR](https://jmlr.org/tmlr/acceptance-criteria.html)) across two aspects that are key to reproducibility in machine learning:

- *Choice of hyper-parameters*. In most experiments, the default values in the original paper's code are used, and it is not clear that those are the values used to generate the results of the original paper (the values of $\lambda_1=1$ and $\lambda_2=0.5$ do not seem to match those reported in Section 5.3 of the original paper). Furthermore, the authors do not systematically study the robustness of the method to the choice of hyper-parameters.

- *Error bars and choice of seed*. The original paper does not say whether they evaluate their approach on a single or multiple runs of their approach, and does not provide any error bars. This is not good practice and I would expect a reproducibility study to improve this evaluation by evaluating on multiple random runs and producing error bars. Instead, this study simply re-used the same random seed as (presumably) used in the original paper.

**Resubmission Of Major Revision:**

The authors may consider submitting a major revision at a later time.